# Spatiotemporal Modeling of the Smart City Residents' Activity with Multi-Agent Systems

**Robert Olszewski [1] , Piotr Pałka [2],\*, Agnieszka Turek [1], Bogna Kietlińska [3], Tadeusz Płatkowski [4] and Marek Borkowski [2]**

[1]   Faculty of Geodesy and Cartography, Warsaw University of Technology, Warsaw 00-661, Poland; robert.olszewski@pw.edu.pl (R.O.); agnieszka.turek@pw.edu.pl (A.T.)
[2]   Faculty of Electronics and Information Technology, Warsaw University of Technology, Warsaw 00-665, Poland; M.Borkowski@stud.elka.pw.edu.pl
[3]   Institute of Applied Social Sciences, Warsaw University, Warsaw 00-927, Poland; b.kietlinska@uw.edu.pl
[4]   Institute of Applied Mathematics, Warsaw University, Warsaw 02-097, Poland; tplatk@mimuw.edu.pl
\*   Correspondence: p.palka@ia.pw.edu.pl; Tel.: +48-691-241-565

**Abstract:** The article proposes the concept of modeling that uses multi-agent systems of mutual interactions between city residents as well as interactions between residents and spatial objects. Adopting this perspective means treating residents, as well as buildings or other spatial objects, as distinct agents that exchange multifaceted packages of information in a dynamic and non-linear way. The exchanged information may be reinforced or diminished during the process, which may result in changing the social activity of the residents. Utilizing Latour's actor–network theory, the authors developed a model for studying the relationship between demographic and social factors, and the diversified spatial arrangement and the structure of a city. This concept was used to model the level of residents' trust spatiotemporally and, indirectly, to study the level of social (geo)participation in a smart city. The devised system, whose test implementation as an agent-based system was done in the GAMA: agent-based, spatially explicit, modeling and simulation platform, was tested on both model and real data. The results obtained for the model city and the capital of Poland, Warsaw, indicate the significant and interdisciplinary analytical and scientific potential of the authorial methodology in the domain of geospatial science, geospatial data models with multi-agent systems, spatial planning, and applied social sciences.

**Keywords:** multi-agent systems; smart city development; spatiotemporal modeling; actor–network theory; geoparticipation; social interactions

## 1. Introduction

Many different disciplines use multi-agent systems as a research tool. One of them is the analysis of social relations in the city, as well as the interaction between residents and spatial objects (the background of the research). The open problem to address is an analysis of various factors that influence changes in the level of residents' social engagement in the process of social participation; above all, changes in the level of mutual trust amongst residents and their trust in social institutions. The multi-agent system (in further parts of the paper, the authors use MAS abbreviation) that models the process of changing the social engagement of residents, proposed by the authors of the article, is the main contribution to the scientific research integrating applied social sciences, geoinformation technologies, and multi-agent systems.

In his 2007 article [1], Michael F. Goodchild introduces the concept of social assembling of spatial information by users identified as specific human agents. Implementing this idea brought about the

rapid development of the so-called VGI (volunteered geographic information), manifested by, e.g., social (crowdsourcing) creation of Open Street Map by over 3 million active users around the world.

The purpose of this article is to develop the concept of smart cities' residents as active urban sensors represented by agents in the MAS. Consequently, city residents are considered elements of a specific geospatial multi-agent system. Mutual interactions of the residents, between the residents, as well as the impact the "human agents" have on spatial objects in the long-term influences the activeness of the residents.

It should be emphasized that for this concept, the crucial assumption is that mutual interactions between residents and spatial objects are characteristic of a complex multi-agent system in a smart city. The non-linear exchange of packages of information between individual elements of the system, under special conditions, leads to reinforcing information and "activating" residents in the process of social participation. It is assumed that the residents (represented by agents) also interact with different features of the city, which tend to modify their trust.

In recent decades, the crisis of participatory democracy has been particularly severe in urban centers and areas subject to urbanization. Its outcome is the weakening of the sense that the residents have a real impact on co-creating the vision for a city's development, revitalization of the neglected districts, or spatial order. When interpreting the term "the right to the city," David Harvey [2] emphasizes that it is not only the right to access its resources, but also the right to decide jointly on the direction of the city's development. A smart and sustainable city engages all of its residents in the most critical decision-making processes, making the process of creating spatial order more social, and encouraging the development of participatory and deliberative democracy. An increase in mutual trust among residents and their trust in public institutions are of crucial importance to stimulating the activity. Therefore, the goal of the authors is to model complex social interactions between the urban residents and to establish the level of trust, sense of identity, willingness to participate in the social (geo)participation processes, and the dynamics of changes over time. The devised model has been tested not only on the example of the "model city," but also on the real urban agglomeration of Warsaw, Poland.

The authors intend to develop a concept of a multi-agent decision support system that makes use of game theory, multi-agent systems, and market programming models to support the "weakest link" of the asymmetric urban network's triangle of "municipal authorities" (politics), "business" (urban developers, industrial investments, etc.), and "residents". So that the atomized individuals are transformed into a cooperative urban community, it is necessary to use available means of electronic communication, information and communication technologies (ICT), and geoinformation tools, as well as to revive the Athenian ideas of the (urban) agora that decides on the city through the process of social debate. Such an asymmetry is the basic rule for producing the majority of urban relations described in Latour's actor–network theory [3]. According to this theory, the basic element of all the networks is an actant, meaning a factor influencing all the other factors. In this article, the social sciences' idea of an actant will be synonymous with an agent. This term will apply both to a "human agent" that is a resident characterized by a vector of information specific to his/her age, education, place of residence and work, general health condition, base level of social activity, trust, and so on, as well as a spatial object (district, office, park, and so on) that influences the inhabitants. These factors influence each other, forming a system of actors and networks.

On the basis of this theory, the authors of the article developed a model of interaction between the sensors–actants (both residents and spatial objects that form the urban tissue), which enables the simulation of social interaction processes as well as, indirectly, participatory democracy and the process of social participation of smart cities' residents. The critical element of this process is stimulating the civic activity of the citizens by increasing the level of trust, both in people and institutions.

The mutual trust of residents, as well as the level of the citizens' trust in institutions (e.g., municipal and security authorities, planners, educational institutions, healthcare system, and so forth), is particularly important with regards to implementing the idea of a smart city. While technological

development is important to ensure the effectiveness of the process, the development of social interactions among the residents and their joint decision-making on the vision for the city's development is crucial. As stressed in [4], cities of tomorrow need to adopt a holistic model of sustainable urban development. A city is smart when public issues are solved using information and communications technology (ICT) and there is involvement of various types of stakeholders acting in partnership with the municipal authorities (see [5]). The implementation of a smart city is strongly related to the process of social (geo)participation—according to PAS 180: 2014 (Publicly Available Specification under the British Standards Institution), this process means an "effective integration of physical, digital and human systems in the built environment to deliver a sustainable, prosperous and inclusive future for its citizens." Also, the new urban agenda stresses the need to empower all individuals and communities and to promote and broaden inclusive platforms that allow full and meaningful participation in the decision-making and planning process (see [6]). A city can be considered a smart one when it, in parallel, invests in technology and human capital to actively promote sustainable economic development and high quality of life (e.g., it enables natural resources management through civic participation).

The contributions of the paper are (i) development of sociological concepts: Bruno Latour's actor–network theory [3], Edward T. Hall's social distances [7], Erving Goffman's social interactions [8], related to sensors–actants which enables the simulation of social interaction; (ii) implementation of the model using multi-agent methodology [9] in GAMA toolset environment; (iii) spatiotemporal and sociological development of concepts of two smart cities and their implementation in GAMA; and (iv) an illustration of the model's operation in typical situations occurring in cities. The use of these approaches made it possible to model the social activity of residents in a dynamic and non-linear way, as well as to conduct spatiotemporal analysis, and create geospatial data models.

The article consists of six sections. After a short introduction (Section 1) the authors describe related works regarding the analyzed issue and motivate the choice of the methods used (Section 2). Subsequently, the authors discuss the research methodology (Section 3). In this section, the authors discuss the actor–network theory (Section 3.1), which is the basis of our model, as well as the method of city modeling used in this article (Section 3.2), then go to the description of city modeling using agents (Section 3.3) and interactions in which these agents participate (Section 3.4). The next section describes in detail, validates, and calibrates the model city (Section 4). The authors use two scenarios for this purpose: Terra incognita (Section 4.1) and Old Factory revitalization (Section 4.2). This is followed by an analysis of the spatiotemporal model of the city of Warsaw (Section 5) and subsequent sections analyze three scenarios: Parade Square (Section 5.1), "Mordor" on Domaniewska Street (Section 5.2), and Miasteczko Wilanów (Section 5.3). The work ends with a discussion and conclusions (Section 6).

## 2. Related Works

The interdisciplinary nature of the research undertaken by the authors of this article requires referencing numerous concepts and methods derived from urban planning, spatial planning, sociology, spatial science, as well as mathematics or computer science.

There have been numerous attempts at developing appropriate tools using modern technologies, e.g., geospatial multi-agent system design and integration, agent-based systems, machine learning, data mining, augmented reality, virtual reality, or 3D models, to ensure effective participation of citizens in urban and territorial development decision-making with a game theoretical treatment (see [10,11]). In [12], authors predict that the widespread presence of smartphones will soon mean that citizens will be treated as a network of sensors that the city will use for continuous development. The use of the concept of a personal digital assistant to support a smart-city citizen, which is most often run on smartphones, is described in the paper [13]. Authors propose a software prototype of a personal digital assistant 2.0, which, based on soft computing methods and cognitive computing, improves calendar and mobility management in smart cities. On the other hand, there are many publications on the analysis of social behavior in urban environments by using multi-agent systems or agent-based

models. Malleson, in [14], emphasizes the need to combine big data and agent-based modeling tools to analyze a smart city. Karmakharm and Richmond, in [15], include a description of the pedestrian behavior simulation in the event of a threat in the public space. Meanwhile, Sandhu et al., in [16], present the implementation of a model, based on intelligent agents, for controlling streetlights in a smart city. In their study [17], Olszewski, Pałka, and Turek analyze the problem of traffic jams in the office district with regards to car-sharing. Their agent-based model simulates the socio-economic behavior of the employees of the so-called Mordor of Warsaw.

Geo-sensing enables context-aware analyses of physical and social phenomena. Moreover, context-aware analysis can potentially enable a more holistic understanding of spatio-temporal processes [18], where authors discuss the possibilities of integrating spatiotemporal contextual information with human and technical sensor information. Among different types of sensors used to collect such kinds of information, they mention in situ sensors, technical remote sensors, and human agents, discussed by Sagl, Resch, and Blaschke [19]. Resch, in [20], defines human agent data as human-generated measurements. He distinguishes the situation in which humans generates data (subjective observations) and humans that carry "ambient sensors" to measure external parameters. In the literature, there were also attempts made for interpreting data acquired by a "human agent", who uses an interactive location-based service (iLBS) (e.g., to sense cultural-historic facts in the landscape) (see [21]).

Cellular automata (CA) can be used to simulate urban dynamics and land-use changes effectively. Several authors performed simulations of urban development and land-use changes using GIS-based cellular automata (see [22–25]). Li et al., in [26], indicate that using parallel computation techniques can significantly improve the performance of the large-scale urban simulation. Agent-based models are applied to increase the intelligence and flexibility of planning support systems. Saarloos et al., in [27], developed a framework in which an agent organization consists of three types of agents: "interface agents" to improve the user–system interaction; "tool agents" to support the use and management of models; and "domain agents" to provide access to specialized knowledge.

Imottesjo and Kain, in [28], developed a prototype mobile augmented reality (MAR) tool, Urban CoBuilder. The application facilitates participative planning of urban space to increase bottom-up and multi-stakeholder inclusion. Yan Zhang, in [29], prototyped CityMatrix, which is an evidence-based urban decision support system, augmented by artificial intelligence (AI) techniques, including machine learning simulation predictions and optimization of search algorithms. Zhang investigated the strength of these technologies to augment the ability to make better urban decisions. Allen, Regenbrecht, and Abbott, in [30], investigated a smartphone-based augmented reality architecture as a tool for aiding public participation in urban planning by developing a prototype system, which showed 3D virtual representations of proposed architectural designs visualized on top of the existing real-world architecture. The authors investigated whether using a smartphone augmented reality system increases the willingness of the public to participate and the perceived participation in urban planning.

Jing and Hai-xing [31] built a support vector machine (SVM) model to predict the trends of coordinated development. The authors compared the method with an artificial neural network, decision tree, logistic regression, and naïve Bayesian classifier regarding the urban ecosystem coordinated development prediction for the Guanzhong urban agglomeration.

Ultsch, Kretschmer, and Behnisch, in [32], used techniques of machine learning and data mining to discover comprehensible and useful structures in the multivariate municipality data. As Behnisch and Ultsch in [33] indicate, "Urban Data Mining represents a methodological approach that discovers logical, mathematical and partly complex descriptions of urban patterns and regularities inside statistical data".

In the conducted research, multi-agent systems were adopted as a tool for modeling and simulation. It enabled the implementation of the assumptions behind the actor–network theory for modeling social processes in the urban space of a smart city. A multi-agent system (MAS or a "self-organized system") is a computerized system composed of multiple interacting intelligent agents (see [9,34]). Multi-agent systems can solve problems that are difficult or impossible for an individual agent or a monolithic system to solve. The primary assumption of MAS is communication amongst the agents and their

autonomy. The notion of agents, as currently used in urban simulation models, is a kind of automaton that mimics the behavior of urban agents in a predetermined way. Portugali, in [35], describes a CogCity (cognitive city) as an urban simulation model that explicitly incorporates in its structure the role of three cognitive processes that typify the behavior of human agents: Information compression, cognitive mapping, and categorization. Moreover, the model CogCity demonstrates the possibility and usefulness of agent-based and cellular automata urban simulation model, which combines top-down and bottom-up processes in one model. Projects from MIT's SENSEable City Lab foster the vision of the real-time city by providing 'a feedback loop between people, their actions, and the city'.

The issue of public opinion formation is the subject of studies conducted by Deffuant, Amblard, and Weisbuch in [36], and Hegselmann and Krause in [37]. These authors consider the issue of social opinion formation through consensus, polarization, and fragmentation. The article investigates various models for the dynamics of continuous opinions by analytical methods as well as by computer simulations. Consequently, the rapid development of advanced technologies (IoT, wearable computing, etc.) forces the process of connecting real-world objects like buildings, roads, household appliances, and human bodies to the Internet via sensors and microprocessor chips that record and transmit data such as sound waves, temperature, movement, and other variables. This supports the development of smart citizens (see [38]).

In [39], Jacobs points to correlation between the urban form and the urban performance, e.g., the quality of life, vibrancy, and safety. Yan Zhang, in [29], takes it a step further and shows the correlation between the urban form and multiple aspects the urban performance. The 17 defined indexes represent 17 aspects of the urban performance of a city district, grouped into four high-level indexes: Density, diversity, proximity, and energy.

Sociological theories have been a vital source of inspiration for the authors of this article. Source studies include the theory of social impact developed by Nowak–Latané, which describes the interaction among members of large groups and the stabilization of opinions in groups [40,41]. Nowak, Szamrej, and Latané in [42] argue that "[modeling] the change of attitudes in a population resulting from the interactive, reciprocal, and recursive operation Bibb Latané's theory of social impact, which specifies principles underlying how individuals are affected by their social environment". Also, the dramaturgical Goffman's theory [8] and Latour's actor–network theory (ANT) [3] have been of crucial importance for the conducted research.

According to Latour's actor–network theory (ANT), an actant, an advanced sensor, is the basic component of all networks; it is a factor influencing other factors. ANT is a theoretical and methodological approach to social theory where everything in the social and natural worlds exists in constantly shifting networks of relationship [3]. It posits that nothing exists outside those relationships. All the factors involved in a social situation are on the same level, and thus, there are no external social forces beyond what and how the network participants are interacting at present. Thus, objects, ideas, processes, and any other relevant factors are seen as just as necessary in creating social situations as humans. Latour distinguishes two types of ties between the actants: Active and passive. The result of an active one is not typical and depends on the mediation between the mediators, meaning that the result is uncertain and variable. In the case of ties between mediations, the situation is stable, and the translation proceeds in a predictable and predetermined manner.

The authors have also been inspired by the studies that take into account the opinion formation model with a "strong leader" (see [43,44]), meaning a leader who significantly influences the molding and modifying of the opinions and attitudes of the residents.

## 3. Research Methodology

The authors of this article see the relationship between the structure of a city, spatial order, the way residents live, and the level of their social activity. The issue of information asymmetry is of crucial importance for modeling these relationships (see [45]).

### 3.1. Actor–Network Theory

Depending on the context of the study, a given actant may be divided into a more complex actor–network order (e.g., a city may be analyzed as a system of relations between buildings, districts, authorities, residents, road infrastructure, and so on). The strength of the influence of individual actants—which is a result of various factors, such as the level of mutual trust or the identification with a given place or space—determines the strength of the relations. Furthermore, a number of such forces may impact a single actant at a given moment, each of the forces with suitable power, which is often asymmetrical in relation to others. This lack of symmetry, or an uneven distribution of forces and influences, manifests itself in almost every dimension of the polysemous creation that is a city. Starting from the right to determine, through the levels of capital (social, economic, cultural, and symbolic) of the city's individual users, to planning and urban solutions that may result in, among other things, ghettoization or spatial exclusion of specific groups of residents. However, one may assume that, firstly, an uneven flow of information between particular actants, often conditioned by the previously mentioned level of social trust, underlies each of the urban asymmetries. Secondly, individual asymmetries overlap and form relations with other asymmetry systems, resulting in the production of additional, now much more intricate, networks of mutual influences and interactions. The level of trust is crucial for the activity and social participation of the citizens.

The underlying assumption of the authors in relation to the concept of geospatial multi-agent system design is that modeling of social interactions is a non-linear generalized regression. It is, therefore, assumed that:

- Only selected factors (out of an infinite number of factors) influencing the level of social activity of residents are analyzed in the model. The advantage of this approach is the opportunity to use quantitative models; the disadvantage is the omission of the factors described in the sociological theories of a qualitative nature.
- Modeling the time changes of the sensor–actor system means complex and multiple interactions of individual sensors, which requires considering the iterative approach and simulating long-term processes.
- Following the idea of citizens as sensors, smart city residents are "human agents" that, during multiple interactions, exchange packets of information all the while modifying the parameters that characterize individual elements of the system (actants). According to the actor–network theory, achieving a certain level of parameter "trust" causes the social activation of individual residents.
- Every "human agent" is an autonomous entity with individual goals and information (also conflicting). Every agent strives to achieve its own goals, unattainable without interacting with others. This approach is consistent with the agent programming paradigm, which provided the basis for modeling the system using multi-agent methodology. Also, it is assumed that "human agents" are dynamic objects as interactions change their attributes.

In their research, the authors investigate how the trust of residents change with time: Both the level of mutual trust and the trust in social institutions, which then stimulates the growth of social involvement and social (geo)participation. The level of trust and its changes depend on factors such as, among others, place of residence, type of work, time spent in public transport or public facilities, theatres, as well as the types of building development or the openness of space, and so on. Changing all of the parameters for a population of hundreds of thousands of people requires the use of parallel computing in multi-agent systems and numerical simulations covering millions of calculation epochs, which model decades of a city's functioning. The base level of the residents' identities, the intrinsic idea of deliberative democracy, the so-called strong leaders in the local community, as well as the specific genius loci of the city are all crucial in the process of changing the level of trust and involvement of residents.

### 3.2. Modeling of the Smart City

The crucial element of studying the development of a city and the way the urban network works using the (broadly defined) game theory is determining whether the knowledge of individual players (advanced sensors) is symmetrical or if there is an informational asymmetry. Multi-agent systems are one of the tools used for modeling the game theory and the theory of market mechanisms.

The models use elements of game theory, emergence, sociology, and multi-agent systems. The model assumes that agents, representing the residents of a city (sensors), move around the city and interact with each other. It takes place during every act of verbal or non-verbal communication. This process involves mutual decoding and simultaneous interpretation of the meaning of symbols used by the other party in communication. Interactions influence and change agent's trust in other residents. The information asymmetry phenomenon is easily modeled in a multi-agent system where every agent (an autonomous software element), while making decisions based on private information and interacting with the remaining agents, has a piece of information whose level may be varied. The simulation is divided into small time quanta (e.g., 15 min), during which the residents interact with each other.

The city is modeled by a system of roads on which the agents move, a set of buildings, including stand-alone and multi-family residential buildings, factories, office buildings, offices, health clinics, schools, and museums; green areas, i.e., boulevards, parks, and water reservoirs. The city is also divided into districts distinguished by a set of general features, which characterize both the district and the people in the district, e.g., the office district is characterized by a significant share of office buildings, and the people in it are blue-collar workers.

### 3.3. Citizen Modeling—Agents

An agent models a city resident and has a set of features that reflect its social character:

- Age, gender, marital status, and number of children.
- Trust in other residents, a number in the range <0,1> that determines to what extent a resident trusts other people. It is not a pejorative trait; it does not mean naivety, but faith in the capabilities of others.
- Trust in institutions, a number in the range <0,1> that determines to what extent a citizen trusts institutions; that is, governmental agencies, healthcare, or educational institutions.
- Altruism, a number in the range <0,1> that determines to what extent a person wants to work for the society and engage in the social life for the sake of common interests.
- Education, a number in the range <0,1> that determines the degree of education.
- Life satisfaction, a number in the range <0,1> that determines to what extent a citizen is satisfied with her/his life.
- Wealth, a number in the range <0,1> that determines the material status of a citizen.
- Identity, a number in the range <0,1> that determines the emotional connection of the citizen with the city or district in which s/he lives.

Besides the features above, each agent is assigned to a place in which s/he lives (a residential building) and a workplace (an office building or a factory). During the simulation, the agents are moving around the city according to the daily rhythm. Residents navigate the city along a network of roads; eventually, they can go to a demonstration.

Demonstrations take place in the so-called attractors—places that attract social interest and provoke extreme emotions. One such attractor is the Palace of Culture and Science in Warsaw, as its demolition is a continuing matter of dispute. A demonstration causes a clash of extreme emotions of the participants and often results in a change of stance regarding the fate of a given attractor.

The devised model assumes that from Monday to Friday, each agent (human agent) leaves for work in the morning (06:00 to 08:00). An agent stays at work for eight hours and then returns home. Some people go to a governmental agency during work hours (10:00 to 12:00) or to a doctor (09:00 to 10:00). After work, and on the weekends, some of the agents leave the city (18:00 to 23:00). Agents meet and interact when traveling, walking around the city, arriving at work, or places of entertainment.

In addition to ordinary residents, there are also the so-called leaders: Social activists who want to impact on the society so that, following the idea of Václav Havel's "The Power of the Powerless" essay [46], many of the weak gain influence over the molding of the urban fabric through mutual interactions. Leaders are close to controversial events. Positive leaders are those with extreme and affirmative trust in people; they have a beneficial influence over the social trust in others. Negative leaders place extreme and negative trust in people, thereby lowering the trust of people with whom they interact towards others.

### 3.4. Agents' Interactions

Meetings between residents take place when they are in the same buildings and when the agents are moving around the city. During these meetings, interactions occur, affecting the change of the agents' characteristics. Such interactions are a symptom of human spatial behavior, resulting from the social distances described by Hall in [7]. Generalized Tobler's first law serves as the starting point for modeling the processes of urban interactions in accordance with Latour's actor–network theory. It states that "everything is related to everything else, but near things are more related than distant things." In the modeled process, objects (actors) influencing each other are actants (sensors) understood as the residents of the city, buildings, spatial arrangement, the dominating function of a given city district, and so on. It is important to emphasize that the "nearness" of the actants here (in this context, the residents) can mean not only the distance in the geographical space, but also the similarity of characteristics, shared interests and views, or social media connection. There is also an assumption that interactions may occur through social media or online interactions, without the need for agents to meet physically. Thus, what is taken into account is not the physical distance between the agents (Euclidean distance in the space of a city), but the distance of the social network, which depends on the educational or age difference. The following interactions, two of them defined by Goffman in [8], latter proposed by the authors, are considered:

- Focused interaction: Influences an agent located in a personal distance (45 to 120 cm). It is assumed that the residents participating in focused interaction will pursue their own goals. An interaction occurs through observation, listening, speaking, and acting. The roles of the interaction parties are strictly defined. The model assumes that a leader should be one of the interaction parties.
- Symbolic interaction: Influences an agent located in a personal and social distance (45 to 360 cm). Symbolic interactionism discussed in [47,48] has also been crucial for the research. It is a sociological perspective dealing with the study of interactions taking place as a result of symbols and gestures. Symbolic interactionism is based on the analysis of the mutual interactions processes, understood as the exchange of symbolic meanings. An exchange takes place between conscious partners who are continually interpreting the situation. The interaction between individuals consists of sending, receiving, and interpreting symbols. The devised model assumes that this interaction occurs between all agents.
- Social media interaction: Influences an agent located in a social net distance (age difference <0.05 or educational difference <0.05). It is a process of exchanging opinions and comments among the users of the social network while reading posts, watching movies or images, listening to sound recordings, and conversations.

Additionally, there is an assumption that the mere presence of a citizen in a given district or a building affects her/his characteristics and, in particular, her/his trust in other people. Being in an office, healthcare facility, at a workplace, school, or in an industrial or office district reduces trust (temporarily or permanently). On the other hand, being in a park, museum, on the boulevards, or in a historic or recreational district increases trust (temporarily or permanently). These changes are called the location influence.

The characteristic analyzed by the authors is the change in the level of mutual trust between residents ("human agents") resulting from the multidimensional influence of the actants. The level of their "trust" tends to "equalize" with the interaction of agents representing individual residents,

although it is not an immediate or rapid process. The conducted simulations aim to determine how the long-term change of people's trust (mutual and to institutions) over time affects the level of social participation and, indirectly, the development of open civil society and deliberative democracy.

The model includes three groups of actants (agents) interacting with each other in the city:

1. People (residents);
2. Institutions (governmental agencies, businesses, educational facilities, healthcare);
3. Spatial objects (districts characterized by parks, monuments, rivers, and so on).

In the adopted model, the authors of the study take into account the various modifying functions (see Figure 1) of the trust parameter in the interactions between two agents:

- The "linear" function, which is the basis for the formulation of the others. The assumption is that this function, with the interaction of the agent *i* with the agent *j*, is as follows:

$$trust^i_{People} \, trust^i_{People} + c \cdot \Delta_{trust_{People}} \tag{1}$$

$$ck_{edu} \cdot \left(1 - edu^i\right) + k_{hap} \cdot \left(1 - hap^i\right) + k_{wea} \cdot \left(1 - wea^i\right) + k_{age} \cdot \left(1 - age^i\right) + k \tag{2}$$

$$\Delta_{trust_{People}} \, trust^j_{People} - trust^i_{People} \tag{3}$$

where:

$k_{edu}$—education parameter change modifier;
$k_{hap}$—satisfaction parameter change modifier;
$k_{wea}$—wealth parameter change modifier;
$k_{age}$—age parameter change modifier;
$k$—change modifier.

- The "reinforcing" function differs from the linear function in the manner of calculating parameter $\Delta_{trust_{People}}$ :

$$\Delta_{trust_{People}} \, tan\left(trust^j_{People} - trust^i_{People}\right). \tag{4}$$

- The "diminishing" function also differs from the linear function in the manner of calculating parameter $\Delta_{trust_{People}}$ :

$$\Delta_{trust_{People}} \, tgh\left(trust^j_{People} - trust^i_{People}\right). \tag{5}$$

- A function that uses the fan-idol relation; based on the situation that a person (fan) will imitate another person (idol) when s/he notices a significant similarity of specific features. Then, s/he changes the parameter (trust in people) towards the idol parameter.
- A function that uses the fan-anti-idol relation; based on the situation that a person (fan) will want to distinguish themselves from the other (anti-idol) when s/he notices a significant difference of specific features. Then, s/he changes the parameter (trust in people) in the opposite direction to the anti-idol parameter.

Moreover, it is assumed that the agent's trust that is included in the above dependencies is treated depending on the place where he resides. The trust of the agent staying at the place of residence is increased by 50%, while for the agent staying at the workplace, it is reduced by 20%. In addition, when the agent is in an entertainment location, his trust counts as 10% more. At the same time, the same trust for an agent staying at the governmental agency or health clinic is reduced by 50%. An agent residing in an industrial area (Old Factory or Mordor) also counts as 15% smaller, while when he is in the GreenLand or OldTown districts, it counts as 15% more.

The introduced methodology is used and validated in the following section.

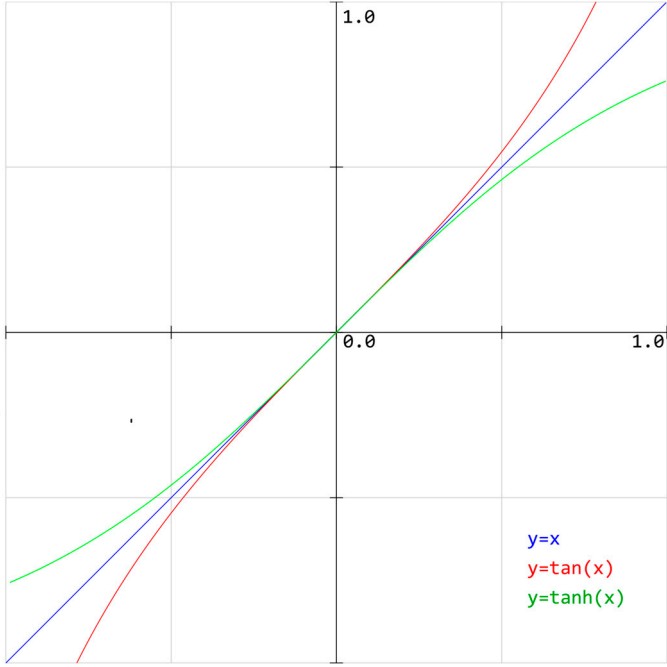

**Figure 1.** Linear (y = x), reinforcing (y = tan(x)), and diminishing (y = tanh(x)) functions.

## 4. Model City

To validate the devised model of actant interaction, the authors developed a "model city" comprising seven hexagonal districts with a dominating type of function and structure (Table 1 and Figure 2). One million residents inhabit the city, as illustrated by the dot distribution map (Figure 3). Each of dots, representing 1000 inhabitants, was implemented into the system as an agent with specific characteristics such as age, education, wealth, marital status, number of children, identity (level of identification with the city), trust in people, and trust in institutions. The variations (standard deviation) of the agent characteristics in each district was assumed at 20% for model data. The parameters for agents were drawn according to the normal distribution.

The ArcGIS ESRI (ArcGIS is the name of software developed by Environmental Systems Research Institute; GIS is abbreviation of Geographic Information System) application was used to develop the source spatial database, enabling the preparation of a set of thematic layers (land cover, buildings, communication routes, districts borders, distribution of residents) as shapefiles. These layers were used to build a multi-agent system in the GAMA simulation platform (see [49,50]). GAMA is a modeling and simulation-development environment for building spatially explicit agent-based simulations (see [50]). It is a multiple-application domain platform using a high-level and intuitive agent-based language. With GAMA, users can undertake most of the activities related to modeling, visualizing, and exploring of the simulations using dedicated tools.

**Table 1.** Model city districts (mean values, in percent); standard deviation is equal to 20%.

| No | Name | Population | Trust to People (%) | Trust to Institutions (%) | Altruism (%) | Education (%) | Happiness (%) | Wealth (%) | Identity (%) | Age |
|----|------|-----------|---------------------|---------------------------|--------------|---------------|---------------|------------|--------------|-----|
| 1 | Greenland | 50,000 | 90 | 80 | 70 | 90 | 100 | 90 | 100 | 60 |
| 2 | City Center | 250,000 | 40 | 60 | 30 | 60 | 60 | 60 | 90 | 50 |
| 3 | Bedroom Suburb | 400,000 | 40 | 50 | 30 | 70 | 50 | 50 | 20 | 40 |
| 4 | Old Town | 50,000 | 80 | 60 | 80 | 80 | 80 | 80 | 100 | 70 |
| 5 | Business District | 100,000 | 50 | 60 | 20 | 70 | 50 | 60 | 20 | 30 |
| 6 | Old Factory District | 150,000 | 30 | 20 | 20 | 20 | 10 | 10 | 40 | 60 |
| 7 | Unspecified Space | 0 | | | | | | | | |

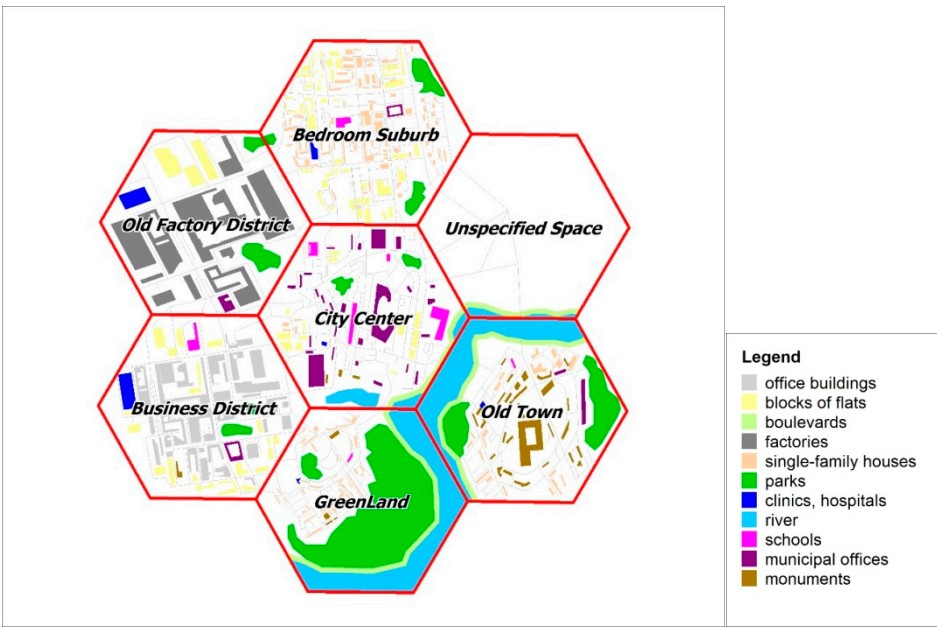

**Figure 2.** Model city.

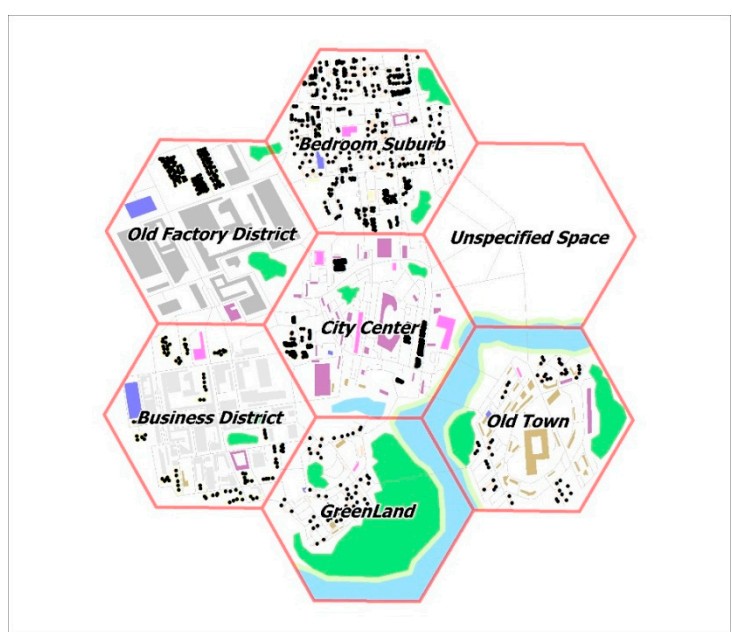

**Figure 3.** Dot distribution map (each dot represents 1000 inhabitants of the model city).

Thanks to the simulations carried out for the model city, it was possible to calibrate the multi-agent system properly, i.e., to determine the value of individual parameters, which then enabled the use of the devised model for a real urban agglomeration area. For example, 3650 iterations used by the authors correspond to a period of 10 years, during which the level of trust of residents changes significantly (and in a measurable way). The authors repeatedly modified the numerical values of particular factors (e.g., changing the level of trust of individual actants resulting from their mutual interactions) so that the parameterization of the model corresponds to the changes observed in the real cities. Because of the iterative calibration of the system, it was possible to determine the parameters of the model adequate for the research of real metropolises.

The analyses made it possible to check the spatial distribution of changes in the level of trust of the residents of particular districts (Figures 2 and 3) in a long-term (decades-long) process. Thanks to the

use of a multi-agent system, it was possible to simulate many years of social processes in computational cycles lasting from several dozen minutes to several hours.

The study also examined the influence of some strong "positive" or "negative" leaders, the impact of the adopted function modifying the traits of the agents (linear, reinforcing, diminishing), as well as specific spatial problems in the model city. In the research, the authors adopted four analytical scenarios:

- City functioning normally;
- The problem of spatial development of the "empty" seventh district;
- The problem of revitalizing a factory going into liquidation;
- The problem of changing the development of the model city's central square (market).

Each of the scenarios is associated with the engagement of a specific group of residents (e.g., the elderly, the less affluent, or the residents of a given part of the city).

Each agent is characterized by the "susceptibility" parameter, which determines the probability of an agent taking part in the social debate, demonstration, or protest. This characteristic is determined on the basis of the agent's other parameters, such as age, education, family, wealth, the level of identification with the city, and so forth. What is key, however, is a given agent's place of residence and the proximity (spatial or social) to the place where a problem, such as the revitalization of a district or the demolition of a controversial town hall, occurs. "Ordinary" agents engage (with a certain probability) in social life after work or on weekends. Only agents representing the "strong leaders" always remain in the conflict places. These agents do not change their level of involvement or trust during the interaction. For positive leaders, it is 1.0, while it is 0.0 for negative ones.

Making use of the devised multi-agent system and the GAMA toolset environment, 3650 computational epochs were carried out. The characteristics of individual agents changed in each iteration because of contacts with individual actants (residents and spatial objects). Using GIS tools, the resulting data were subjected to spatial aggregation analyzing the change in the average level of trust of agents residing in a given district of a model city. First, the authors of the article made calculations, the purpose of which was to check how the particular trust parameter modifying functions works (Figures 4 and 5).

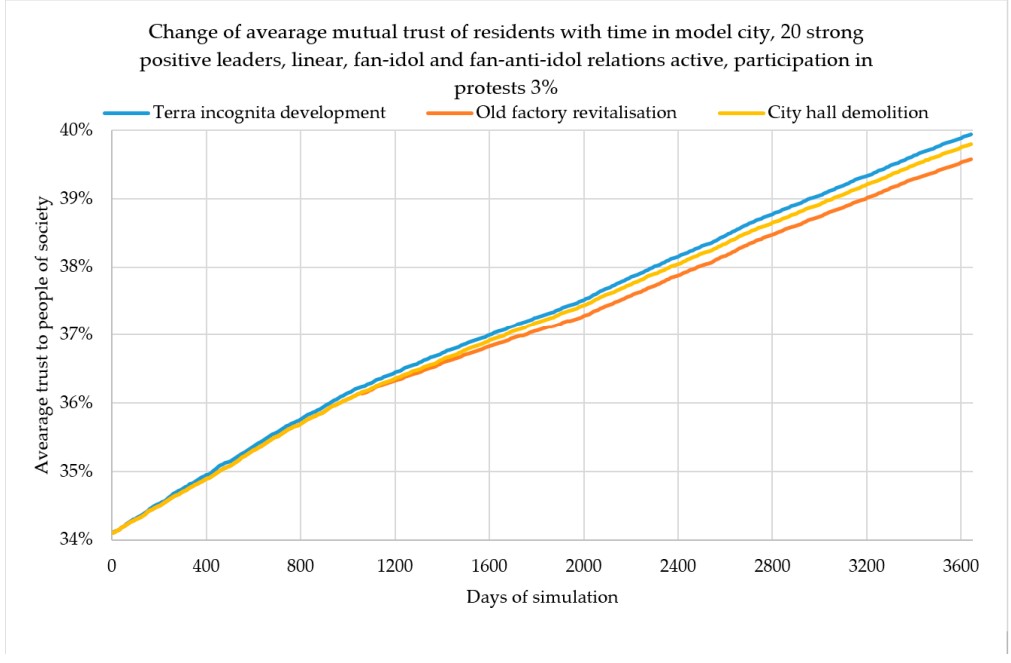

**Figure 4.** The attractor activity and the level of mutual social trust in a model city through a simulated time of 10 years in the presence of 20 strong positive leaders.

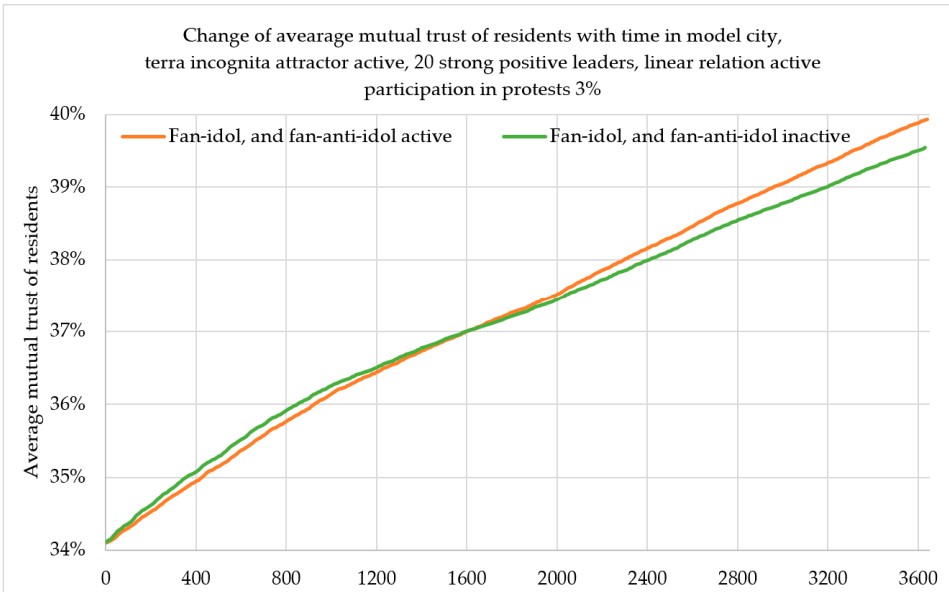

**Figure 5.** Comparison of the fan-idol and fan-anti-idol relations on the level of mutual social trust in a model city through a simulated time of 10 years in the presence of 20 strong positive leaders.

In Figure 4, one can observe the comparison of the attractor's activity and the level of mutual social trust in a model city through a simulated time of 10 years in the presence of 20 strong positive leaders. All plots rise steadily, but with slightly different slopes. Moreover, in the period of attractors' activeness (days 1200–2400), the curves are moving away from each other. The results indicate that there are differences between individual attractors, which results from various geospatial settings.

In Figure 5, one can observe the comparison of the fan-idol and fan-anti-idol relations on the level of mutual social trust in a model city through a simulated time of 10 years in the presence of 20 strong positive leaders. What is interesting is that up to the 1600-ish day of simulation (four years and four months), the inactivity of the fan-idol and fan-anti-idol relations seems to be stronger. However, after that, the activity of the relations begins to be stronger, and at the end, after 10 years, the mutual trust is 0.4% stronger, compared to the inactivity of the relations.

To sum up, the analysis of various attractors and various functions modifying confidence shows us minor differences, which result from the differences characteristic for a given area, rather than differences in the algorithm used.

The variants of analytical simulations implemented are presented below.

### 4.1. Terra Incognita (Unspecified Space)

Research question: Will the district be built in a "closed" way (gated communities) or (because of the increased level of social (geo)participation and trust) in an "open" way? It is of particular interest to the residents from three marked districts, and young, relatively wealthy, and well-educated people.

Figure 6 shows the comparison between the number of strong leaders and the level of mutual social trust in a model city through a simulated time of 10 years, with fan-idol and fan-anti-idol relations inactive. In the absence of strong leaders, the slope of mutual trust decreases over time and the increase in 10 years is 1.12%. For 20 strong positive leaders, trust increases by 5.44% within the simulation, but the slope is changing; during the manifestation, the slope slightly decreases. It can be explained by the concentration of all the positive leaders in one district of the city. However, in the case of 20 strong negative leaders, the authors observe a slight increase in trust in the beginning, but on the 2000-ish day of simulation (around year 5.5), it begins to drop. Ultimately, after 10 years, trust increases by 1.12%, which is the result of accumulating negative opinions during protests. Similar dependencies, with accuracy to value, occur for other attractors. Finally, the existence of both 20 positive and 20 negative

strong leaders results in a different outcome than the non-existence of strong leaders. On the contrary, one can note that the positive leaders have more impact on mutual trust than the negative ones. For this case, the mutual trust after 10 years of simulation increases by 2.94%.

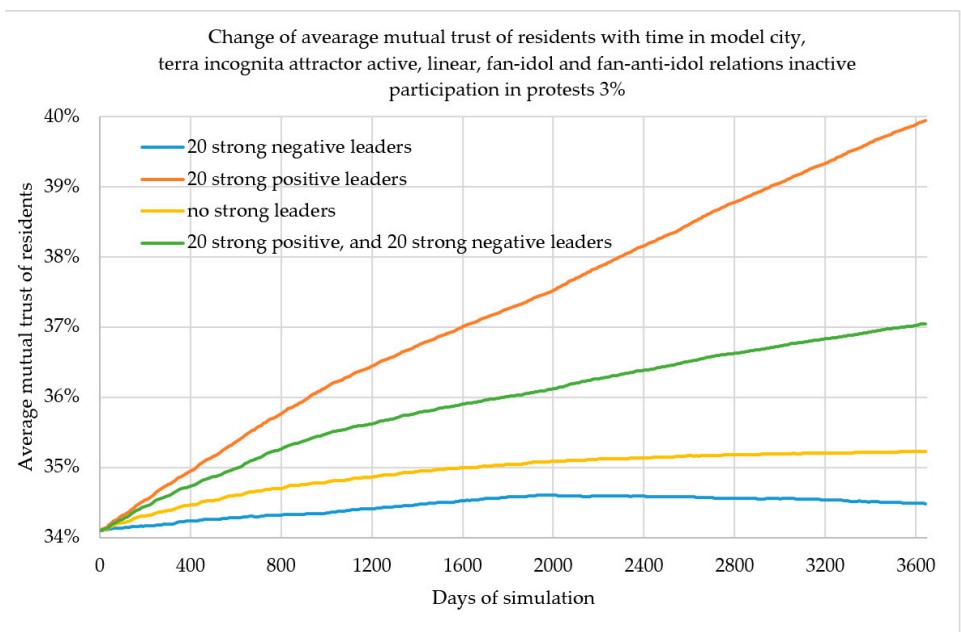

**Figure 6.** Comparison of the number of strong leaders and the level of mutual social trust in a model city through a simulated time of 10 years, with fan-idol and fan-anti-idol relations inactive.

In Figures 7 and 8, one can see that the changes in the level of trust differ significantly in individual districts of the model city. These changes also have different intensity over time. This process depends not only on the number of strong leaders, but also on the characteristics of residents of particular districts and the level of their involvement in the problem of this attractor. Positive leaders influence a slight increase in trust in Bedroom, while negative leaders considerably reduce the level of trust of the residents of this district. For those who are not very interested in the spatial development of the new district (residents of Old Town and GreenLand), the level of trust decreases both in the presence of positive and negative leaders, although with varying intensity.

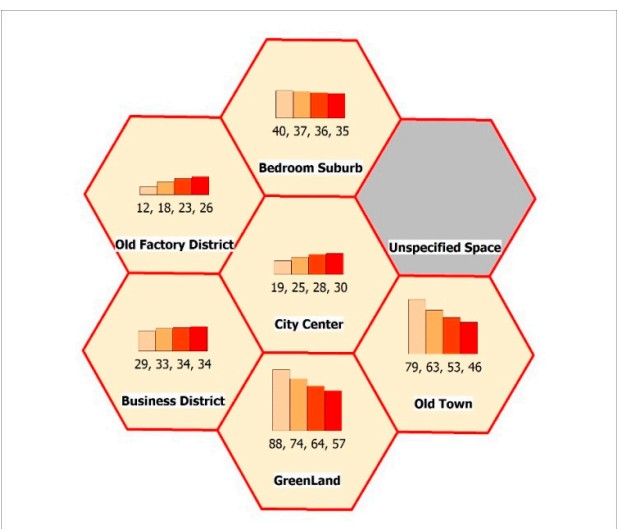

**Figure 7.** Changes in the level of mutual social trust in a model city through a simulated time of 10 years (20 negative leaders in the "Terra incognita" attractor); iteration 0, 1200, 2400, 3650.

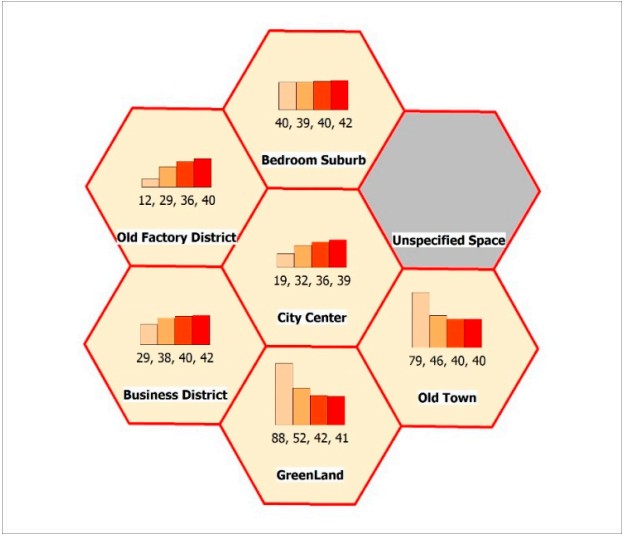

**Figure 8.** Changes in the level of mutual social trust in a model city through a simulated time of 10 years (20 positive leaders in the "Terra incognita" attractor); iteration 0, 1200, 2400, 3650.

### 4.2. Old Factory Revitalization

The revitalization of the building after the Steelworks closes down leads to gentrification—and loft spaces for the wealthy. Conducted studies on spatial and descriptive data characterizing the model city have shown that for the relatively poor employees of the closed down Steelworks and the residents of this region (this district and a spatially close part of the bedroom community, center, and business district), this problem is particularly significant (affects negatively). Figure 9 shows a zone of strong and weak impact of the "Old Factory" attractor.

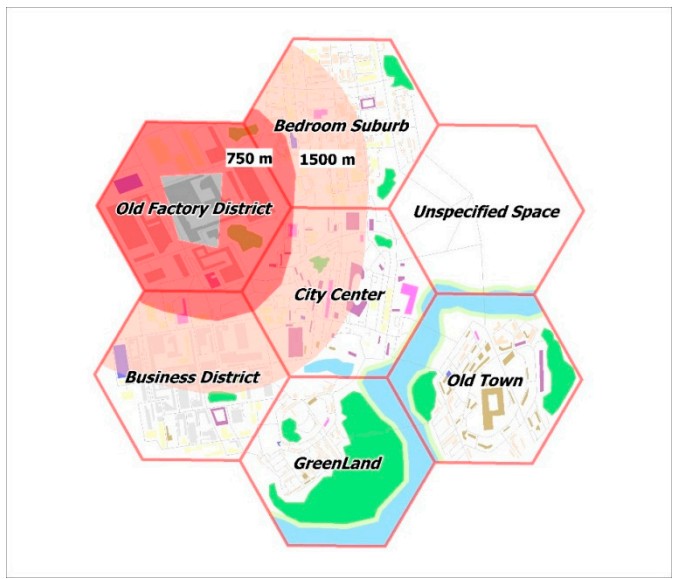

**Figure 9.** Zone of strong and weak impact of the "Old Factory" attractor.

In Figure 10, one can observe that overall increase in trust by 8% does not mean an even increase of confidence in all districts of the model city. The revitalization of the factory and the creation of residential lofts is especially important for the residents of this district and two neighboring ones (business district and city center). This is also important for some residents of the city who work (or have worked) in a liquidated factory, regardless of where they live. For the inhabitants of the bedroom suburbs, the process of revitalization is of little importance to the wealthy inhabitants of the old town

and green land who are not interested in this subject, it even results in lowering the level of social activity. On account of the obtained results, it is possible to state the following:

- A multi-agent system enables simulations of long-term social processes and interactions of the citizens as sensors (both mutual and concerning the urban tissue);
- It is relatively easy to scale the impact of individual factors on the process, which facilitates the development of a model to be used to simulate complex, multi-parameter social processes in real cities;
- The combined use of multi-agent systems, advanced sensors, and GIS tools makes it possible to analyze the interaction of the actants (residents and spatial objects) as well as spatial aggregation and visualization of the results in the form of thematic maps;
- The selection of a spatial database (a digital map of the city) and the spatial distribution of residents and their characteristics make it possible to simulate incredibly complicated processes, e.g., related to spatial development, revitalization, and so on.

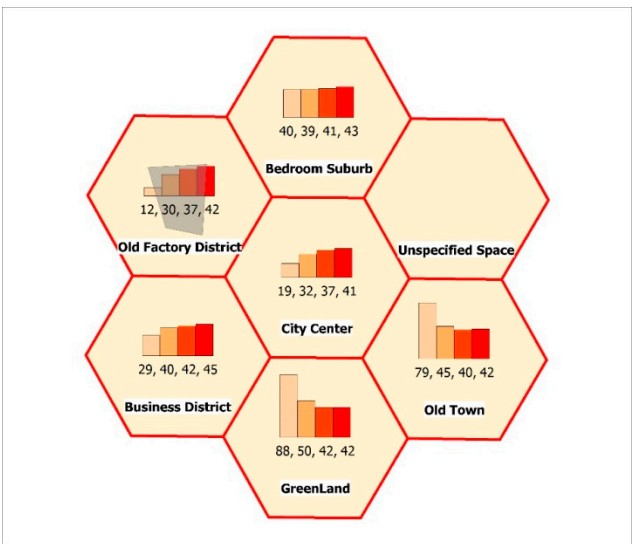

**Figure 10.** Changes in the level of mutual social trust in a model city through a simulated time of 10 years (20 positive leaders in the "Old Factory" attractor); iteration 0, 1200, 2400, 3650.

## 5. Spatiotemporal Modeling of the Warsaw Area, Poland

With an appropriately calibrated model and its implementation in the form of a multi-agent system, the authors attempted to conduct research and simulation on real data. The agglomeration of Warsaw in Poland, with its 1,754,000 inhabitants, was the test object (Figure 11). Spatial data used in the study come from the general geographic database, which contains data at the level of accuracy that is equivalent to analogue maps with a scale of 1:250,000. This study was up-to-date in 2016. As with the model city, the distribution of residents was modeled in the form of a dot distribution map (Figure 12), where each of 1754 dots (agents) represents 1000 inhabitants. The data contain a set of characteristics defining the demographic, social, and cultural features of individual residents (Table 2).

The research [51,52] is examining the level of trust of Poles, both in public institutions and each other. It indicates the direct relationship between trust, civic activity, and the level of education. It turns out that "trust increases civic activity only after reaching or exceeding the threshold of secondary education.". In addition, in 2015, at the request of the City Hall of Warsaw, a study was conducted on the quality of life of the residents of Warsaw districts, in which a set of questions was devoted to trust. Also, although the level of trust in friends and family remains at a level close to 90%, confidence in politicians (17%), journalists (34%), and local authorities in Warsaw (35%) still remains very low.

Analytical experiments were carried out for Warsaw by simulating long-term social processes for three selected test areas (Figure 12):

- The central area of the capital (controversy regarding the spatial development of Parade Square);
- The so-called Mordor (with the problem of extreme street congestion during the day and its forlornness at night);
- The so-called Miasteczko Wilanów (socially associated with a "ghetto" for young and wealthy residents born outside of Warsaw, who have a low level of identification with the city).

**Table 2.** Warsaw (PL) districts (mean values, in percent); standard deviation is equal to 20%. Source: [51].

| No | Name | Population | Trust to people (%) | Trust to institutions (%) | Altruism (%) | Education (%) | Happiness (%) | Wealth (%) | Identity (%) | Age |
|----|------|-----------|---------------------|---------------------------|--------------|---------------|---------------|------------|--------------|-----|
| 1 | Bemowo | 120,000 | 63 | 83 | 20 | 39 | 12 | 30 | 94 | 60 |
| 2 | Bialoleka | 116,000 | 63 | 66 | 22 | 29 | 16 | 70 | 90 | 40 |
| 3 | Bielany | 132,000 | 64 | 80 | 15 | 33 | 11 | 70 | 93 | 70 |
| 4 | Mokotow | 218,000 | 67 | 79 | 11 | 26 | 5 | 80 | 93 | 100 |
| 5 | Ochota | 84,000 | 59 | 75 | 12 | 28 | 5 | 60 | 94 | 80 |
| 6 | PragaPoludnie | 178,000 | 64 | 74 | 19 | 39 | 17 | 60 | 88 | 60 |
| 7 | PragaPolnoc | 66,000 | 61 | 75 | 11 | 22 | 5 | 10 | 86 | 30 |
| 8 | Rembertow | 24,000 | 62 | 72 | 7 | 20 | 4 | 80 | 90 | 60 |
| 9 | Srodmiescie | 118,000 | 63 | 74 | 24 | 41 | 11 | 90 | 93 | 90 |
| 10 | Targowek | 124,000 | 65 | 81 | 19 | 35 | 10 | 40 | 86 | 30 |
| 11 | Ursus | 58,000 | 66 | 84 | 11 | 19 | 3 | 60 | 91 | 70 |
| 12 | Ursynow | 150,000 | 57 | 77 | 23 | 32 | 7 | 80 | 91 | 70 |
| 13 | Wawer | 75,000 | 72 | 79 | 18 | 31 | 2 | 60 | 95 | 70 |
| 14 | Wesola | 25,000 | 69 | 78 | 11 | 24 | 4 | 70 | 96 | 70 |
| 15 | Wilanow | 35,000 | 73 | 76 | 27 | 48 | 14 | 90 | 94 | 90 |
| 16 | Wlochy | 41,000 | 69 | 83 | 15 | 28 | 3 | 70 | 98 | 70 |
| 17 | Wola | 139,000 | 69 | 86 | 25 | 37 | 11 | 50 | 93 | 60 |
| 18 | Zoliborz | 51,000 | 67 | 75 | 17 | 32 | 16 | 80 | 94 | 80 |

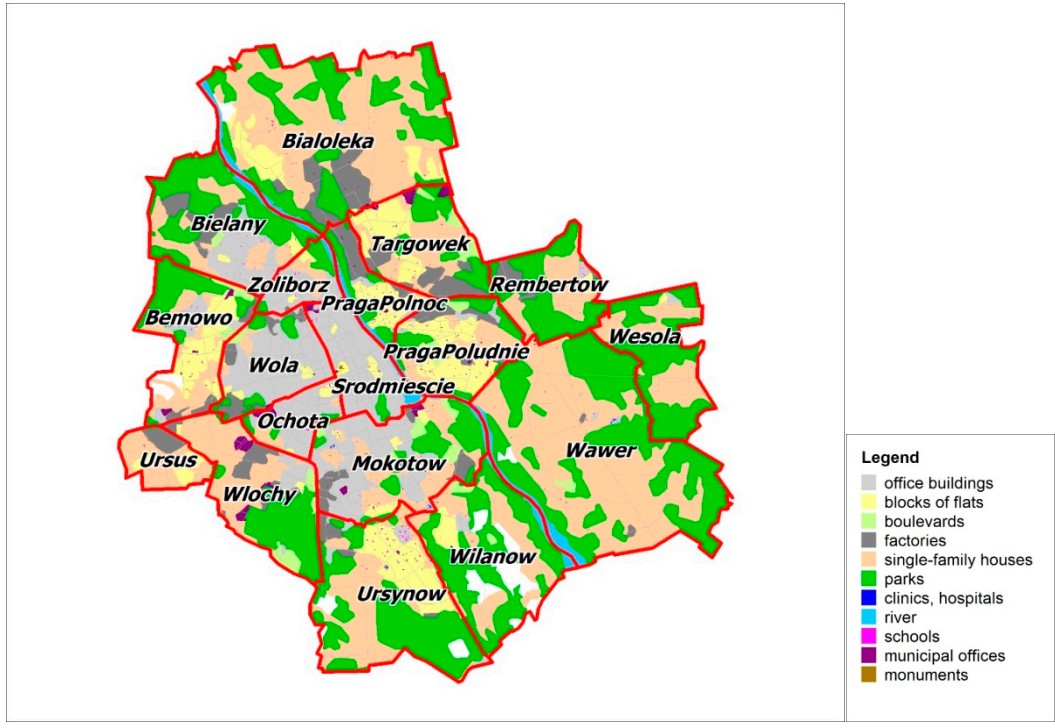

**Figure 11.** Warsaw (PL) and its districts.

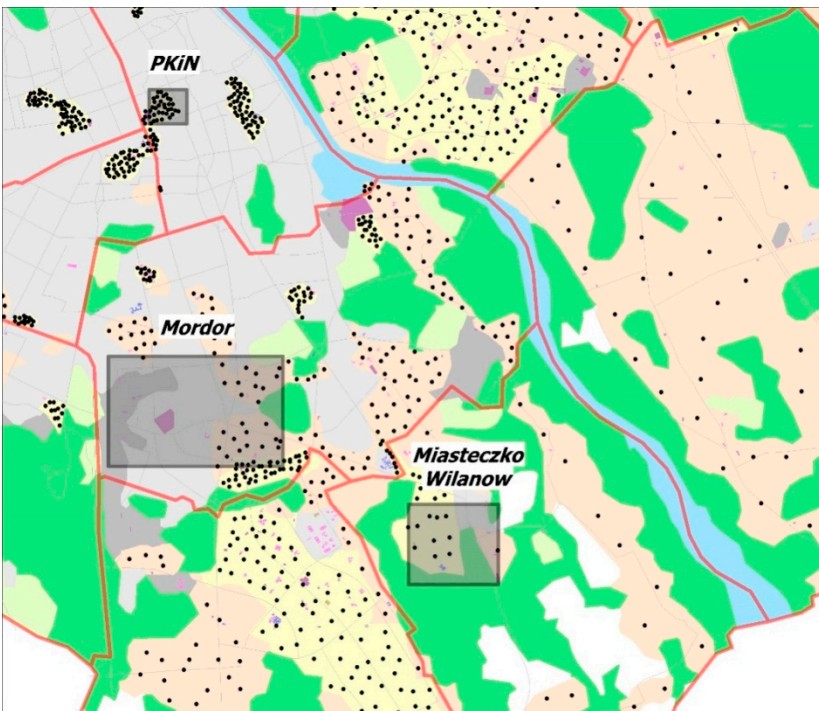

**Figure 12.** Three attractors in Warsaw and dot distribution map.

## 5.1. Parade Square (Plac Defilad)

Together with the Palace of Culture and Science, Parade Square has been a source of controversy for years. The background of this conflict is primarily generational and, thus, historical. The problem lies not only in its development, but also in the possible ways of integrating this space into the city.

The seniors born or living in Warsaw declare low trust and distaste for this area. To this group, this place is connected with the period of communist domination, complicated history, and the Soviet Union. They would like to demolish this space along with the Palace of Culture and Science. This group of people has a very high level of identification with the city and its space, especially with the city center. For the youth and people in their prime (up to around 35 years of age), the Palace remains a symbol of Warsaw and the location of many cultural activities, as well as a place for meetings or dating. Young people, in particular, have been trying to revitalize this area for years. They would like to combine the Palace of Culture and Science with space for everyone, where there is a place for greenery, leisure, and a body of water. A manifestation of this is, for example, the so-called Central Park (https://parkcentralny.pl/), which is a project aimed at creating the Green Heart of Warsaw.

In Figure 13, the change of average mutual trust of residents for Parade Square with 20 negative and 20 positive strong leaders can be observed. The mutual trust decreases by 6.8%, which is the opposite of the case of the Model City, where mutual trust grew in a similar situation (20 positive and 20 negative strong leaders). This process is caused by different demographic characteristics of Warsaw (a real urban agglomeration) and the model city.

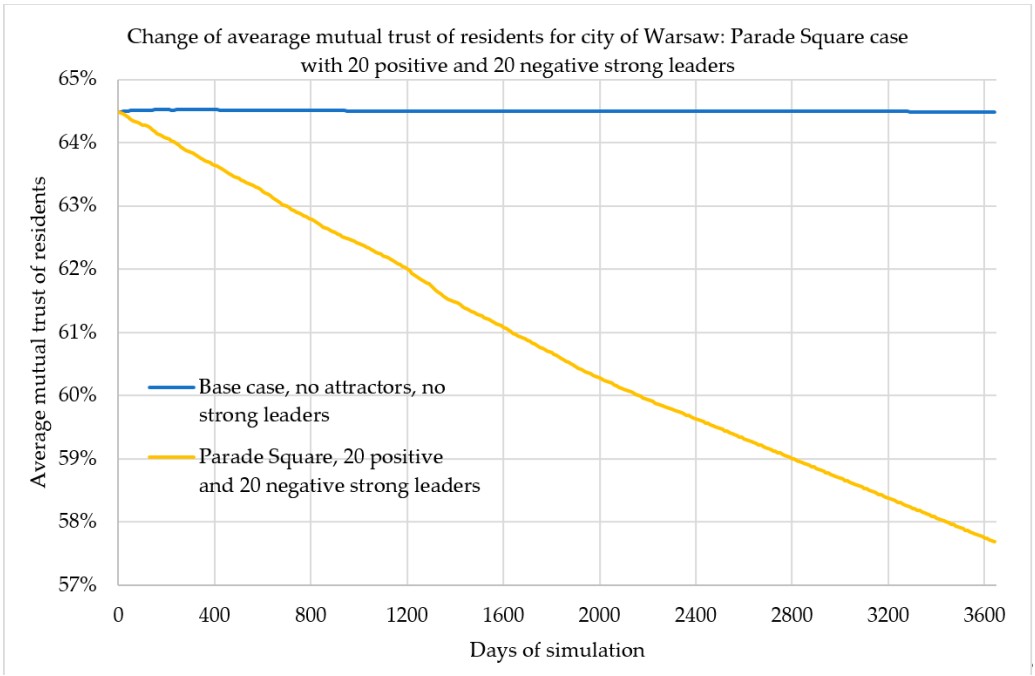

**Figure 13.** Change in the average mutual trust of residents for Parade Square with 20 negative and 20 positive strong leaders.

### 5.2. "Mordor" on Domaniewska Street

In the 1990s, the process of changing this part of the city from one of industrial function to one of a service function began. At the moment, Domaniewska Street is one of the main streets of the largest office complex in Warsaw. The residents of the city have jokingly named it Mordor, which is the dark land from the novels by J.R.R. Tolkien, because of its traffic-related problems.

The problem with this part of the city is the extreme intensification of office space, lack of greenery, and a limited number of parking spaces. A thick line seems to separate the Mordor of Warsaw from the city (for communicational and architectural reasons, but also mentally). This part of Warsaw has always been associated with a low level of identification with the city. After its change from an industrial area to a service one, the problem of low-level identification and trust remained (which also applies to the attitude of officials and corporations). It is a place where one has to be (work), not where one wants to be. Places that appear in Mordor (restaurants, cafés) are there primarily to serve the corporations; there are no cultural or recreational places (except for fitness centers), and so on.

In this part of Warsaw, all actants have a very low level of trust in public transport and, therefore, in officials as well (employees as well as visitors or residents). For people from the outside, Mordor is a place of very low trust and identification (especially for those who live close to this area, e.g., Mokotów, Ursynów, and so on, neighboring districts). People living in the outlying districts are somewhat indifferent to this place. On the other hand, this place "draws" newcomers who decide to buy flats there. As a result, a sense of new urban identity arises, in a way forged as an opposition to the inhabitants of other parts of Warsaw.

The critical problem of this part of Warsaw is the extreme congestion of streets during rush hour. Almost all of the 100,000 employees of corporations in Mordor commute to work using the company car. An increased mutual trust would allow the rationalization of commuting, e.g., by using carpooling (see [17]).

In Figure 14, one can observe the change of average mutual trust of residents for Mordor with 20 strong negative leaders. The constant drop of mutual trust (−16.06%) can be observed (the drop is more than twofold when compared to Parade Square, see Figure 13). It means that for the city of Warsaw, in the case of strong negative leaders, the level of mutual trust is threatened by a significant

reduction. The increase in social trust, and, thus, also the increase in social participation visibility, e.g., through the joint use of company cars leading to increased traffic capacity, therefore requires a strong inspiration of "positive leaders" of social changes in this office area.

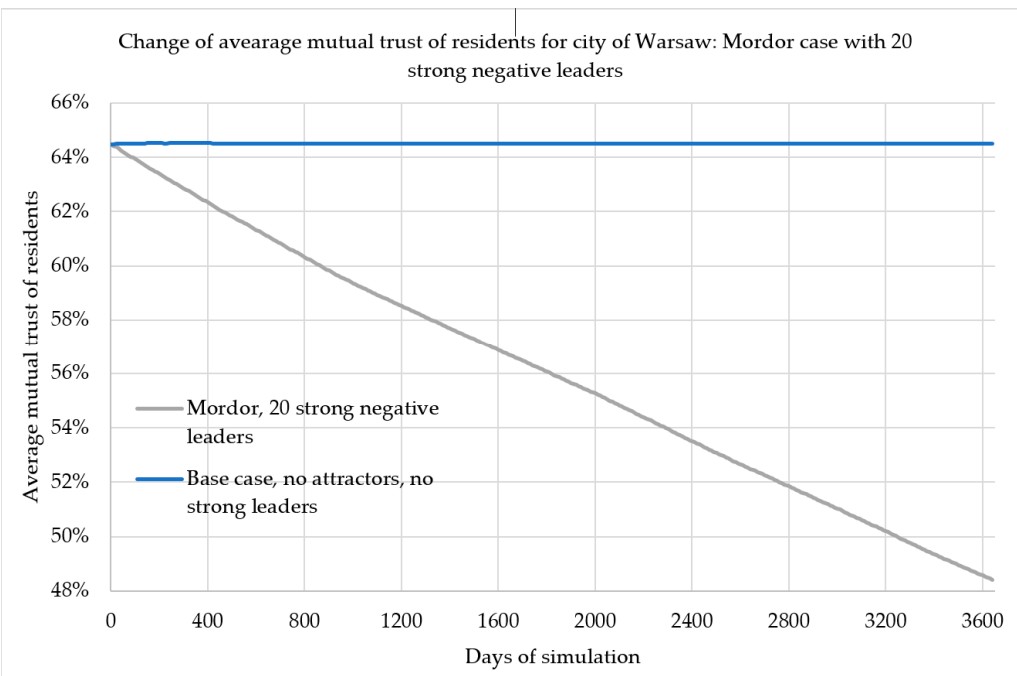

**Figure 14.** Change in the average mutual trust of residents for Mordor with 20 strong negative leaders.

### 5.3. Miasteczko Wilanów

This is a recently built district of the city with gated communities inhabited mainly by newcomers with a low sense of identification with Warsaw. The inhabitants of the "old" Wilanów and other Warsaw districts (apart from Białołęka) do not trust the residents of the "new" Wilanów. The relations within this district are also complicated: Those who live in the prestigious buildings built in the very beginning do not trust those who came to live in buildings of a much lower standard and intended for the middle class (a typical class conflict).

What characterizes this place is the lack of kindergartens, schools, sports fields, and so on, as well as the underdeveloped public transport. Also, there is no broadly understood public space or any green areas.

An increased mutual trust and level of identification would not only facilitate the active social participation of residents, but also improve relations between the residents of various parts of Wilanów (the "old" and the "new) and the other districts of the city.

In Figure 15, one can observe the change of average mutual trust of residents for Wilanów with 20 strong positive leaders. In this case, the mutual trust increases steadily up to the level of 73.48% (an increase of 8.94%). What is interesting, for an analogous number of leaders, the drop for Mordor (see Figure 14) is nearly twice the one for Wilanów, which means that Warsaw is a city endangered by the declining mutual trust, and it is more difficult to increase the trust than to decrease it.

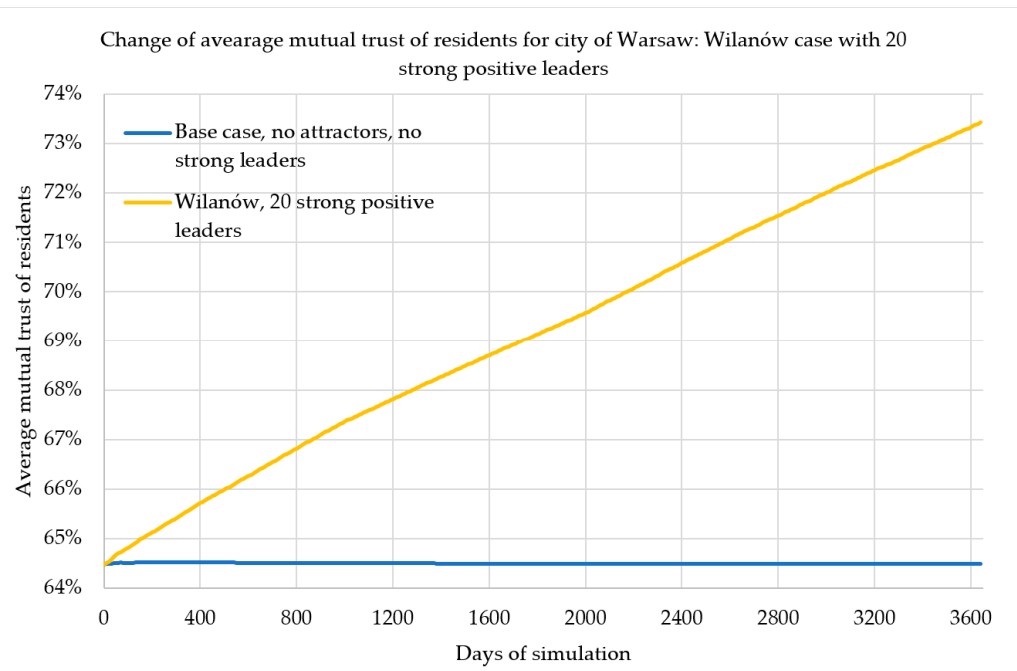

**Figure 15.** Change in the average mutual trust of residents for Wilanów with 20 strong positive leaders.

## 6. Conclusions and Future Work

After completing the simulations and accessing the results, it seems that, thanks to the developed concept and the prototype of a multi-agent information system, which uses spatial data, demographic information, and sociological, mathematical, and urban theories, performing complex geospatial big data analyses, geospatial information extraction, and data mining is possible. Because of the idea of "citizens as sensors" represented by "human agents", game theory, information asymmetry, urban morphology, and multi-agent systems, it was possible to model changes in the residents' activity over decades, even in the cases of agglomerations of hundreds of thousands of residents. Therefore, it is a useful tool not only for conducting urban big data analyses in urban studies, spatial science, or applied social science, but also for shaping the smart cities of the future. The analysis of the results for the model city and Warsaw shows that each city has a "potential for mutual trust" that emerges from the distribution of its buildings, road network and, of course, its inhabitants. This potential of social trust can be substantial, causing an increase in mutual trust, as in the case of the model city. It may also be small, as in the case of Warsaw, where it is difficult to increase mutual trust. However, thanks to strong leaders, it is possible to shape the trust and support the process of increasing the activity of residents—active sensors and their social participation in creating a smart city.

Therefore, the following may be stated:

- The idea of "citizens as sensors," expanded with the elements of actor–network theory and multi-agent systems, facilitates spatiotemporal analysis in geospatial data and spatial knowledge acquisition. Increasing the level of mutual trust between residents and their trust in institutions, as well as the sense of local identity results in increased social activity in the (geo)participation process and co-deciding about the city's development.
- Mathematical models (game theory), social theories, ICT tools, geoinformation technologies, and multi-agent systems constitute a tool for modeling spatiotemporal geoinformation structures and enduring social changes. Because of the use of multi-agent systems to model the asymmetry of social relations, it is possible to analyze changes in the level of residents' activity in the model city and real agglomerations.
- Developing a system for a model city makes it possible to experiment with the value of the factors. As a result, one can examine how an increase or a decrease in social trust affects civic engagement

at a given time. Research on real data enables following the initial situation and then applying the results from the model city to, for example, Warsaw. Consequently, one can determine the factors that directly or indirectly influence the increase of participative activities and then strengthen those elements that require reinforcement.

- The system enables measuring the strength of relations between individual human and non-human agents and then, by the modification of selected elements, to reinforce or diminish them.
- The devised concept of a multi-parameter analytical system and the prototype of a geoinformation tool facilitate not only modeling of the social changes, but also their stimulation. Indirectly, it contributes to the development of a virtual urban agora, deliberative participatory democracy, and the reinforcement of the social (geo)participation processes in the smart communities of smart cities.
- The city is a structure of interdependent networks and relationships. Social trust is one of the elements enabling the effective functioning of society. This is a variable that positively influences the consolidation of democracy and formation of citizenship.
- The results of the research suggest that the context does play a significant role in shaping the effect of social trust on social participation.
- The approach proposed by authors may facilitate constructing more and more holistic models of cities.

It is also worth emphasizing that the developed model can be (after minor modifications) also used in other applications, e.g., stimulating the residents' activity to install photovoltaic panels [53], real estate or multilateral negotiations for building plots in distributed multi-agent environment [54].

As future work, it is worth considering to supplement the proposed multi-agent, agent-based model with a game theoretical treatment, in particular to identify possible social dilemmas, such as e.g., public goods, tragedy of the commons and trust dilemma, and their potential impact on the development of public trust in the considered urban agglomeration of groups of interacting individuals with different interests. The authors of the article also plan to expand the model with elements of gamification between residents to model different ways of social activity of city residents; specifically, it is planned to model the social gamification in a smart city, which is likely to stimulate the installation of the photovoltaic panels. A smart city, understood not as intelligent city infrastructure but as a smart, open geoinformation society, is shaped by the "power of the powerless," which can be reinforced.

The developed model is universal; it can be easily parameterized on the basis of any input data, e.g., social, sociological, or economic. The model will be verified and tested in other agglomerations and different cities. These will include cities characterized by a higher baseline level of trust (Scandinavian cities) and culturally different areas (Singapur, Masdar City).

**Data Availability:** The code of the project in gaml language with included data used to support the findings of this study have been deposited in the git repository: https://gitlab.com/PiotrPowerPalka/smartcitygrowthgama.git.

**Author Contributions:** Conceptualization, R.O. and P.P.; methodology, P.P, R.O., A.T., and B.K.; software, M.B. and P.P.; validation, A.T and T.P.; formal analysis, R.O. and T.P.; investigation, R.O, P.P., and A.T; resources, A.T.; data curation, P.P and B.K.; writing—original draft preparation, R.O, P.P., A.T., and B.K.; writing—review and editing, A.T.; visualization, R.O, A.T, and P.P.; supervision, R.O.; project administration, A.T.

**Funding:** This work was supported by the FabSpace 2.0 project funding from the EU 's Horizon 2020 research and innovation program under the grant agreement No. 693210.

**Conflicts of Interest:** The authors declare that there is no conflict of interest regarding the publication of this paper.

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
