# Peer review of "Spatiotemporal Modeling of the Smart City Residents’ Activity with Multi-Agent Systems"

_applsci, doi:10.3390/app9102059_

Round 1
Reviewer 1 Report
The article is very long and in general exist many terms quite differents which are complicated to relate, however this article it is well explained.
As it is proposed in the text, the aim pursued is the modeling of social interactions between inhabitants of a population and establishing the level of trust, sense of identity, and participation in different social processes, allowing checking the different changes over time.
There are many and multidisciplinary approaches proposed due to the complexity to achieve the objective proposed, and something interesting is that not only the model is deployed under a simulation, it is deployed in a real environment what justifies its validity.
The implementation of the model into the GAMA platform allows us to simulate the proposed spatiotemporal model of an intelligent city by looking at towards the future and therefore gives us a forecast of the behavior of the different social agents all over the time. However, even thought GAMA platform is referenced, in my opinion, it should have an explanation about this platform, and how the model is deployed into it.
Therefore, this model it is a useful not only to analyze large data in urban studies, geosciences or applied social sciences, but also to configure the smart cities of the future.
Author Response
We would like to thank the Editor and Reviewers for a careful and thorough reading of this manuscript and for the thoughtful comments and constructive suggestions, which help to improve the quality of this manuscript. We believe that the revised version can meet the journal publication requirements.
According to the recommendations denoted in the text change in the mode of review, moreover, we have marked a response to comments from reviewers, respectively colors: red Reviewer #1, Reviewer #2 - green, Reviewer #3 - blue.
Our response follows.
Response to comments from Reviewer#1:
Overall comment 1:
As it is proposed in the text, the aim pursued is the modeling of social interactions between inhabitants of a population and establishing the level of trust, sense of identity, and participation in different social processes, allowing checking the different changes over time.
There are many and multidisciplinary approaches proposed due to the complexity to achieve the objective proposed, and something interesting is that not only the model is deployed under a simulation, but it is also deployed in a real environment what justifies its validity.
The implementation of the model into the GAMA platform allows us to simulate the proposed spatiotemporal model of an intelligent city by looking at towards the future and therefore gives us a forecast of the behavior of the different social agents all over the time. However, even thought GAMA platform is referenced, in my opinion, it should have an explanation about this platform, and how the model is deployed into it.
Therefore, this model is useful not only to analyze large data in urban studies, geosciences or applied social sciences but also to configure the smart cities of the future.
Response 1:
We much appreciate the Reviewers very comprehensive review, and positive feedback.
Comment 1.1:
The article is very long and in general exist many terms quite different which are complicated to relate, however this article it is well explained.
Response 1.1:
With regards to the length of the article, it is not easy to shorten the article. We are trying to target both readers that are interested in the comprehensive study and readers that are only interested in subsections. The current form serves both of these needs and we are concerned that if we cut out the small overlaps in the introduction to each section as well as the technical material, then the subsections will not be readable to the reader who is only interested in one or a few sections. Therefore we would like to keep the manuscript largely in its current form. However, as suggested by the reviewer, we have reviewed carefully the entire manuscript and have removed redundancies, as shown in the revised manuscript.
Reviewer 2 Report
In this paper Authors proposed the concept of modeling that uses multi-agent systems of mutual interactions between city residents as well as interactions between residents and spatial objects. The originality the concepts, the significance and the methods are good. The completeness and the organization of manuscript of the paper are good. The organization of the manuscript is good. In my opinion the technical treatment is plausible and free of technical errors. Below I presented some remarks that came to my mind during reading.
Remarks:
1. Line 33: Dot after last keyword is unnecessary.
2. References should be numbered according to the order in which they appear in the text.
3. The paper should be written in an impersonal form.
4. Figures 1, 4, 5, 6, 13, 14 and 15 should be of better quality.
5. References should be prepared in accordance with the Applied Sciences template.
After taking these remarks into account I recommend to accept the paper.
Author Response
We would like to thank the Editor and Reviewers for a careful and thorough reading of this manuscript and for the thoughtful comments and constructive suggestions, which help to improve the quality of this manuscript. We believe that the revised version can meet the journal publication requirements.
According to the recommendations denoted in the text change in the mode of review, moreover, we have marked a response to comments from reviewers, respectively colors: red Reviewer #1, Reviewer #2 - green, Reviewer #3 - blue.
Our response follows.
Response to comments from Reviewer#2:
Overall comment 2:
In this paper Authors proposed the concept of modeling that uses multi-agent systems of mutual interactions between city residents as well as interactions between residents and spatial objects. The originality the concepts, the significance and the methods are good. The completeness and the organization of manuscript of the paper are good. The organization of the manuscript is good. In my opinion the technical treatment is plausible and free of technical errors. Below I presented some remarks that came to my mind during reading.
Response 2:
We much appreciate the Reviewers very comprehensive and detailed review, and positive feedback. We made the following revisions accordingly.
Comment 2.1:
Line 33: Dot after last keyword is unnecessary.
Response 2.1:
We would like to thank the Reviewer for this comment, we deleted the dot.
Comment 2.2:
References should be numbered according to the order in which they appear in the text.
Response 2.2:
Thank you for your valuable notice, indeed due to the continuous work on the article references have been intermingled. According to your comment, we carefully renumbered them according to the order in which they appear in the text.
Comment 2.3:
The paper should be written in an impersonal form.
Response 2.3:
Thank you very much for this attention. In the article, the personal form was replaced by an impersonal form.
Comment 2.4:
Figures 1, 4, 5, 6, 13, 14 and 15 should be of better quality.
Response 2.4:
The quality of the figures is improved.
Comment 2.5:
References should be prepared in accordance with the Applied Sciences template.
Response 2.5:
Thank you for your notice, we have improved the references according to the template.
Reviewer 3 Report
I like the paper, the style and the type of tools and analysis you chose. It is a very good paper, also interesting to open some more discussions on the topic. In Real Estate it can be interesting to see this approach for more researches.
Author Response
We would like to thank the Editor and Reviewers for a careful and thorough reading of this manuscript and for the thoughtful comments and constructive suggestions, which help to improve the quality of this manuscript. We believe that the revised version can meet the journal publication requirements.
According to the recommendations denoted in the text change in the mode of review, moreover, we have marked a response to comments from reviewers, respectively colors: red Reviewer #1, Reviewer #2 - green, Reviewer #3 - blue.
Our response follows.
Response to comments from Reviewer#3:
Overall comment 3:
I like the paper, the style and the type of tools and analysis you chose. It is a very good paper, also interesting to open some more discussions on the topic.
Response 3:
We appreciate the positive feedback from the reviewer.
Comment 3.1:
In Real Estate it can be interesting to see this approach for more researches.
Response 3.1:
The authors of the article thank you very much for this attention. The issue of the use of multi-agent systems can be analyzed in the field of real estate as well as photovoltaic installations. In conclusion, a fragment was added to the use of MAS in real estate. We are looking for a real estate case, we are open for cooperation.
“It is also worth emphasizing that the developed model can be (after minor modifications) also used in other applications, e.g. stimulating the residents' activity to install photovoltaic panels [53], real estate or multilateral negotiations for building plots in distributed multi-agent environment [54].”